# Impact of uncertainties of unbound $^{10}$Li on the ground state of two-neutron halo $^{11}$Li

**Jagjit Singh[1]⋆ and Wataru Horiuchi[2]**

**1** Research Center for Nuclear Physics (RCNP), Osaka University, Ibaraki 567-0047, Japan
**2** Department of Physics, Hokkaido University, Sapporo, 060-0810 Japan

⋆ jsingh@rcnp.osaka-u.ac.jp

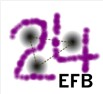

## Abstract

**Recently, the energy spectrum of $^{10}$Li was measured upto 4.6 MeV, via one-neutron transfer reaction $d(^9$Li, $p)^{10}$Li. Considering the ambiguities on the $^{10}$Li continuum spectrum with reference to new data, we report the configuration mixing in the ground state of the two-neutron halo nucleus $^{11}$Li for two different choices of the $^9$Li $+ n$ potential. For the present study, we employ a three-body (core $+ n + n$) structure model developed for describing the two-neutron halo system by explicit coupling of unbound continuum states of the subsystem (core $+ n$), and discuss the two-neutron correlations in the ground state of $^{11}$Li.**

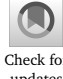

## 1 Introduction

The light dripline nuclei lying away from the strip of stability, have gained prodigious attention of the nuclear physics community over the past few decades and a significant progress has been made both on experimental and theoretical sides to understand their exotic nature [1]. The one of the eye-catching phenomenon in some light dripline nuclei is the formation of halo, which is linked to the small binding energy of one or two valence nucleons [2,3]. Particularly two-neutron ($2n$) halo systems, consisting of a core and two weakly bound valence neutrons, demand a three-body description with proper treatment of continuum. The stability of such three-body (core+$n$+$n$) system is linked to the continuum spectrum of the two-body (core+$n$) subsystem. In this context, to explore the sensitivity of choice of a core+$n$ potential with the configuration mixing in the ground state of three-body systems (core $+ n + n$), we will discuss the results of the $2n$-halo $^{11}$Li.

Although $^{11}$Li is the first observed two-neutron halo four decades ago [3]. Since then a lot of experimental and theoretical studies have been reported on structure of the $^{11}$Li. In order to understand the $^{11}$Li structure, the information over low-lying spectrum of $^{10}$Li is needed as a

fundamental ingredient of three-body calculations. However, the $^{10}$Li structure was studied by various techniques such as fragmentation [4], $^{11}$Li$(p, d)^{10}$Li transfer reaction at TRIUMF [5], multi-neutron transfer [6] and pion absorption reactions [7]. Maximum of these studies report the low-lying $p_{1/2}$ neutron resonance with peak lying in the range of 500-700 keV. Also few of these studies reported the presence of $s$-wave virtual state close to the threshold with a scattering length in the range from $-20$ to $-30$ fm [4] and not much information is available on neutron $d$-wave.

Recently, the $^{10}$Li structure was investigated via $d(^9$Li, $p)^{10}$Li, one-neutron transfer reaction. This study reported $^{10}$Li energy spectrum up to 4.6 MeV, with the existence of $p_{1/2}$ resonance at $0.45 \pm 0.03$ MeV along with other two high lying structures at 1.5 and 2.9 MeV [8]. Also the role of $^{10}$Li resonances is investigated in the halo structure of $^{11}$Li via $^{11}$Li$(p, d)^{10}$Li transfer reaction at TRIUMF [5] and at the same facility the first conclusive evidence of a dipole resonance in $^{11}$Li having an isoscalar character has been reported [9,10]. In view of these new measurements and ambiguities over the experimental data, we aim to explore the sensitivity of the $^9$Li $+ n$ potential with the configuration mixing in the ground state of of three-body system ($^9$Li $+ n + n$).

For this study, we use a three-body (core+$n$+$n$) structure model, developed for studying the weakly-bound ground and low-lying continuum states of Borromean systems sitting at the edge of neutron dripline [11]. In our approach, we start from the solution of the unbound subsystem (core+$n$) and the two-particle basis is constructed by explicit coupling of the two single-particle continuum wave functions. Initially, it was tested for the lightest $2n$-halo $^6$He [12,13], heaviest known $2n$-halo $^{22}$C [14] and $2n$-unbound $^{26}$O [15] and has been successful in explaining the ground-state properties and the electric-dipole and quadrupole responses.

In this contribution, Sec. 2 briefly describes the formulation of our three-body structure model. In Sec. 3 we analyze the subsystem $^{10}$Li and fix the two different sets for $^9$Li $+ n$ potential, consistent with available experimental information. Section 4 presents our results for the three-body system, $^9$Li $+ n + n$. Summary is made in Sec. 5.

## 2 Model Formulation

The three-body wave function for the $^9$Li $+ n + n$ system is specified by the Hamiltonian

$$H = -\frac{\hbar^2}{2\mu} \sum_{i=1}^{2} \nabla_i^2 + \sum_{i=1}^{2} V_{\text{core}+n}(\vec{r}_i) + V_{12}(\vec{r}_1, \vec{r}_2), \tag{1}$$

where $\mu = A_c m_N/(A_c + 1)$ is the reduced mass, and $m_N$ and $A_c = 9$ are the nucleon mass and mass number of the core nucleus, respectively. $V_{\text{core}+n}$ is the core-neutron potential and $V_{12}$ is $n$-$n$ potential. The neutron single-particle unbound $s$-, $p$-, and $d$-wave continuum states of the subsystem ($^{10}$Li) are calculated in a simple shell model picture for different continuum energy $E_C$ by using the Dirac-delta normalization and are checked with a more refined phase-shift analysis. Each single-particle continuum wave function of $^{10}$Li is given by

$$\phi_{\ell j m}(\vec{r}, E_C) = R_{\ell j}(r, E_C)[Y_\ell(\Omega) \times \chi_{1/2}]_m^{(j)}. \tag{2}$$

We use the mid-point method to discretize the continuum. The convergence of the results will be checked with the continuum energy cut $E_{\text{cut}}$ and $\Delta E$. These core+$n$ continuum wave functions are used to construct the two-particle $^{11}$Li states by proper angular momentum couplings and taking contribution from different configurations. The combined tensor product of these two continuum states is given by

$$\psi_{JM}(\vec{r}_1, \vec{r}_2) = [\phi_{\ell_1 j_1}(\vec{r}_1, E_{C1}) \times \phi_{\ell_2 j_2}(\vec{r}_2, E_{C2})]_M^{(J)}. \tag{3}$$

We use a density-dependent (DD) contact-delta pairing interaction [16], given by

$$V_{12} = \delta(\vec{r}_1 - \vec{r}_2)\left(v_0 + \frac{v_\rho}{1 + \exp[(r_1 - R_\rho)/a_\rho]}\right). \tag{4}$$

The first term in Eq. (4) with $v_0$ simulates the free $n$-$n$ interaction, which is characterized by its strength and the second term in Eq. (4) represents density-dependent part of the interaction. The strengths $v_0$ and $v_\rho$ are scaled with the $\Delta E$ by following relation from Ref. [14]. The $v_\rho$ is the parameter which will be fixed to reproduce the ground-state energy. For a detailed formulation and calculation procedure one can refer to Refs. [11–13,17].

## 3 Two-body unbound subsystem (core $+ n$)

The investigation of the two-body (core $+ n$) subsystem is crucial in understanding the three-body system (core $+ n + n$). The interaction of the core with the valence neutron ($n$) plays a fundamental role in the binding mechanism of the three-body system. The elementary concern over the choice of a core$+n$ potential is the ambiguities in the experimental information about the core $+ n$ system. We employ the following core $+ n$ potential with standard choice of spin-orbit interaction,

$$V_{\text{core}+n} = \left(-V_0^\ell + V_{\ell s}\vec{\ell}\cdot\vec{s}\frac{1}{r}\frac{d}{dr}\right)\frac{1}{1 + \exp\left(\frac{r - R_c}{a}\right)}, \tag{5}$$

where $R_c = r_0 A_c^{\frac{1}{3}}$ with $r_0$ and $a$ are the radius and diffuseness parameter of the Woods-Saxon potential. The values of $r_0 = 1.27$ fm and $a = 0.67$ fm are adopted from Refs. [16,18].

Table 1: Parameter sets of the core-$n$ potential for $\ell = 0, 1, 2$ states of a $^9$Li$+n$ system. The possible resonances with resonance energy $E_R$ and decay width $\Gamma$ in MeV are also tabulated.

| Set | $\ell j$ | $V_0^\ell$ (MeV) | $V_{\ell s}$ (MeV) | $E_R$ (MeV) | $\Gamma$ (MeV) |
|-----|----------|------------------|--------------------|-------------|----------------|
|     | $s_{1/2}$ | 50.50 | – | – | – |
| A   | $p_{1/2}$ | 40.00 | 21.02 | 0.46 | 0.36 |
|     | $d_{5/2}$ | 47.50 | 21.02 | 2.98 | 1.39 |
|     | $s_{1/2}$ | 47.50 | – | – | – |
| B   | $p_{1/2}$ | 40.00 | 21.02 | 0.46 | 0.36 |
|     | $d_{5/2}$ | 47.50 | 21.02 | 2.98 | 1.39 |

For the present calculations we ignore the spin of the core $^9$Li. The neutron number 6 is assumed for the neutron core configuration given by $(0s_{1/2})^2(0p_{3/2})^4$. The four valence neutron continuum orbits, i.e., $p_{1/2}$, $d_{5/2}$, $s_{1/2}$ and $d_{3/2}$ are considered in the present calculations for $^{10}$Li. $^{10}$Li is interesting in the sense that it shows inversion of $s_{1/2}$ and $p_{1/2}$ levels.

The scattering length of the virtual $s$-state, position and width of low-lying $p$-resonance along with higher lying $\ell = 2$ resonance vary from experiment to experiment. In the view of the new experimental measurements [5,8], we use two different potential sets for core $+ n$ potential, which are tabulated in Table 1. The only difference between our two sets A and B is we use different $s$-wave depth ($V_0^0$), leading to different scattering length of the $s_{1/2}$ virtual state, which further effect the $s$-wave component in ground state of $^{11}$Li. In our set A the $s$-wave potential is deep enough to increase the $s$-component dominance in the ground state of

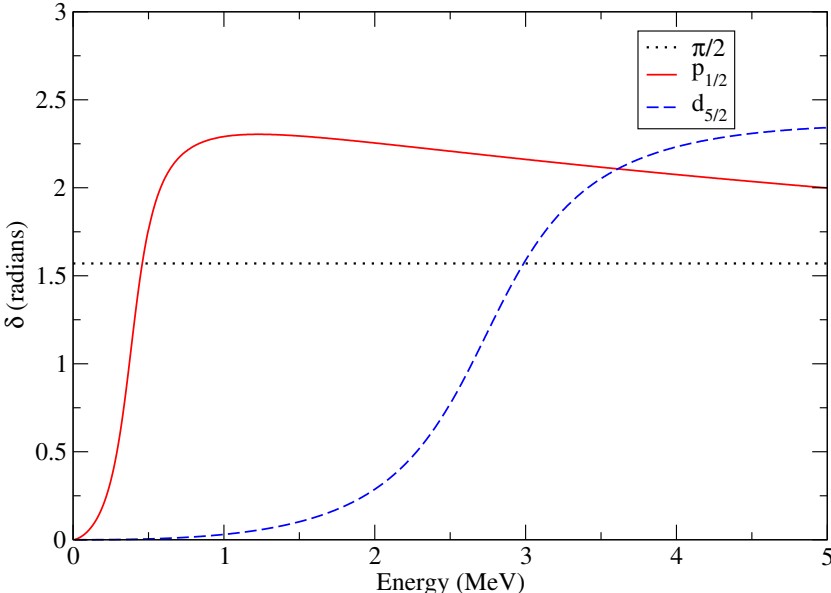

Figure 1: $^9$Li+$n$ phase shifts for $1/2^-$ and $5/2^+$ states corresponding to core+$n$ potential tabulated in Table. 1

$^{11}$Li in comparison to set B. Our both sets reproduces the observed $p_{1/2}$ resonance at 0.45 MeV consistent with Ref. [8] and the $d_{5/2}$ resonance, that lies at higher energy around 2.98 MeV, this position is consistent with the high-lying structure of $^{10}$Li reported in Ref. [8]. The phase-shifts corresponding to these resonances are shown in Fig. 1. Similar potentials are used also in Refs. [16, 18].

## 4  Results and Discussions

The three-body model with two non-interacting particles in the above single-particle levels of $^{10}$Li, produces different parity states, when two neutrons are placed in different unbound orbits mentioned in Sec. 3 (for details see Table. 2). The corresponding oscillatory single-particle continuum wave functions for $s_{1/2}$, $p_{1/2}$, $d_{5/2}$, and $d_{3/2}$ states are plotted in Fig. 2. The four configurations $(s_{1/2})^2$, $(p_{1/2})^2$, $(d_{5/2})^2$, $(d_{3/2})^2$ couple to $J^\pi = 0^+$ for $^{11}$Li.

Table 2: Possible configurations of $^{11}$Li arising from two neutrons in $s$-, $p$- and $d$-orbitals.

|           | $s_{1/2}$ | $p_{1/2}$ | $d_{3/2}$ | $d_{5/2}$ |
|-----------|-----------|-----------|-----------|-----------|
| $s_{1/2}$ | $0^+$     | $0^-,1^-$ | $1^+,2^+$ | $2^+,3^+$ |
| $p_{1/2}$ |           | $0^+$     | $1^-,2^-$ | $2^-,3^-$ |
| $d_{3/2}$ |           |           | $0^+,2^+$ | $1^+,2^+,3^+,4^+$ |
| $d_{5/2}$ |           |           |           | $0^+,2^+,4^+$ |

The continuum single-particle wavefunctions are calculated with energies from 0.0 to 5.0 MeV and normalized to a delta for the $spd$-states of $^{10}$Li on a radial grid which varies from 0.1 to 100.0 fm with the $^9$Li+$n$ potential discussed in Sec. 3. In the three-body calculations, along with the core+$n$ potential the other important ingredient is the $n$-$n$ interaction. We use the DD contact-delta pairing interaction, with the only adjustable parameter being $v_\rho$. The

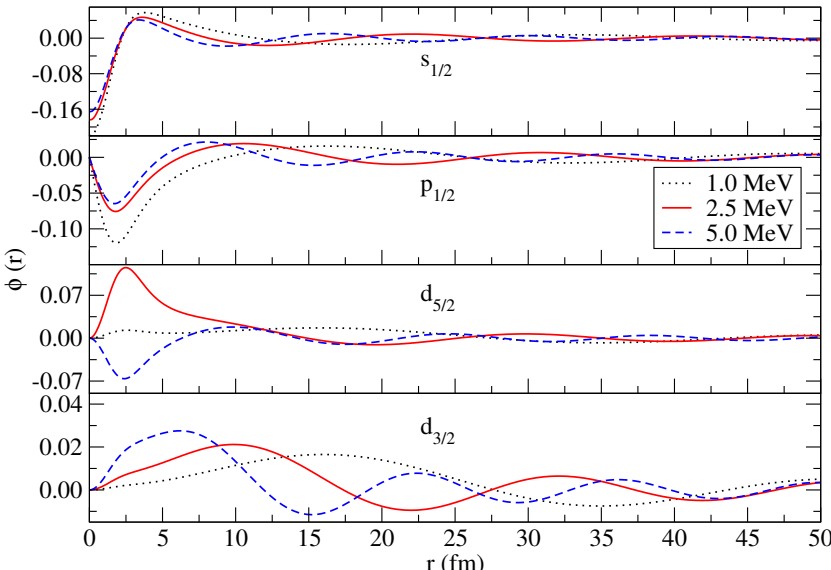

Figure 2: $^9$Li+$n$ continuum waves as a function of radial variable for continuum energies 1, 2.5 and 5 MeV, respectively.

two particle states are formed using mid-point method with an energy spacing of $2.0, 0.5, 0.25$ and $0.1$ MeV corresponding to block basis dimensions of $N = 5, 10, 20$ and $50$, respectively, and the matrix elements of the pairing interaction are calculated. In Fig. 3, the eigenspectrum for $J = 0^+$ case is presented and from figure it is clear that with increase in basis dimensions the superflous bound states moves into the continuum. The biggest adopted basis size gives a fairly dense continuum in the region of interest.

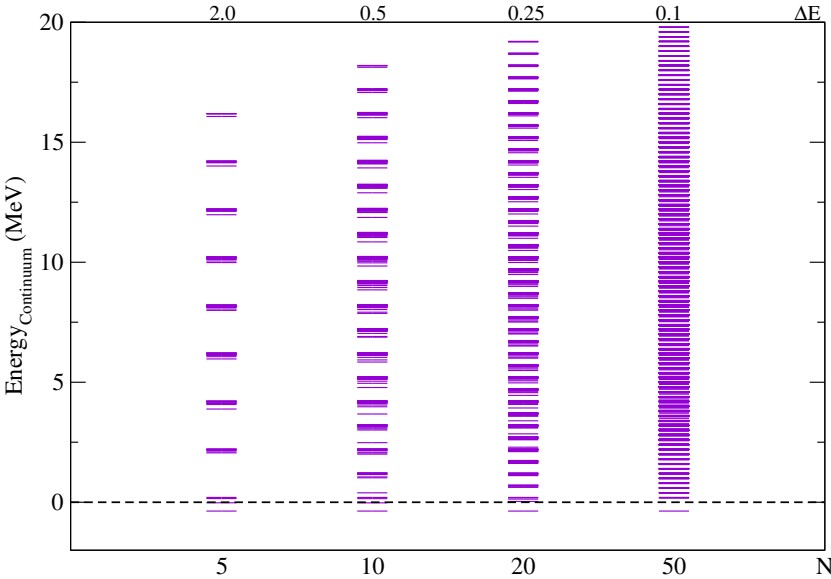

Figure 3: Eigenspectrum of the interacting two-particle case for $J = 0^+$ for increasing basis dimensions, $N$. The parameter of pairing interaction $v_\rho$, has been adjusted each time to reproduce the two-neutron separation energy ($S_{2n}$).

In the DD contact-delta pairing interaction (defined by Eq. (4)), the strength of the DI part is given as $v_0 = 2\pi^2 \frac{\hbar^2}{m_N} \frac{2a_{nn}}{\pi - 2k_c a_{nn}}$, where $a_{nn}$ is the scattering length for the free neutron-neutron

scattering and $k_c$ is related to the cutoff energy, $e_c$, as $k_c = \sqrt{\frac{m_N e_c}{\hbar^2}}$. We use $a_{nn} = -15$ fm and $e_c = 30$ MeV [16], which leads to $v_0 = 857.2$ MeV fm$^3$. For the parameters of the DD part, we determine them so as to reproduce the two-neutron separation energy of $^{11}$Li, $S_{2n} = -0.369$ MeV [19]. The values of the parameters that we employ are $R_\rho = 1.25 \times A_c^{\frac{1}{3}}$ ($A_c = 9$) and $v_\rho = 862.5$ and $861.75$ MeV fm$^3$ for set A and B, respectively.

We report the percentage configuration mixing in the ground state of $^{11}$Li in Table 3. We found that for Set A for which $V_0^0$ is deeper shows dominance of $(s_{1/2})^2$ configuration in the ground state leading to formation of $s$-neutron halo. Whereas for Set B for which $V_0^0$ is shallower shows dominance of $(p_{1/2})^2$ configuration in the ground state leading to formation of $p$-neutron halo. The preliminary numbers for calculated matter radii with these potential sets are 3.53 and 3.24 fm for Set A and B, respectively. These results of configuration mixing and matter radii are consistent with the results of Refs. [16,20] for $^{11}$Li. The detailed investigation of the configuration mixing with inclusion of core spin is in progress.

Table 3: Components of the ground state of $^{11}$Li in %, with the model parameters energy cut, $E_{cut} = 5$ MeV and bin size, $\Delta E = 0.1$ MeV. The core+$n$ potential used are tabulated in Table 1.

| Set | $lj$ | Present work | Reference [20] |
|---|---|---|---|
| A | $(s_{1/2})^2$ | 55.5 | 64.0 |
|   | $(p_{1/2})^2$ | 33.1 | 30.0 |
|   | $(d_{5/2})^2$ | 7.1 | 3.0 |
| B | $(s_{1/2})^2$ | 24.5 | 27.0 |
|   | $(p_{1/2})^2$ | 59.6 | 67.0 |
|   | $(d_{5/2})^2$ | 9.1 | 3.0 |

The two particle density of $^{11}$Li as a function of two radial coordinates, $r_1$ and $r_2$, for valence neutrons, and the angle between them, $\theta_{12}$ in the LS-coupling scheme is given by

$$\rho(r_1, r_2, \theta_{12}) = \rho^{S=0}(r_1, r_2, \theta_{12}) + \rho^{S=1}(r_1, r_2, \theta_{12}). \tag{6}$$

The explicit expression for $S = 0$ component is given by [16, 21]

$$\rho^{S=0}(r_1, r_2, \theta_{12}) = \frac{1}{8\pi} \sum_L \sum_{\ell, j} \sum_{\ell', j'} \frac{\hat{\ell}\hat{\ell}'\hat{L}}{\sqrt{4\pi}} \begin{pmatrix} \ell & \ell' & L \\ 0 & 0 & 0 \end{pmatrix}^2 (-1)^{\ell+\ell'} \sqrt{\frac{2j+1}{2\ell+1}} \sqrt{\frac{2j'+1}{2\ell'+1}}$$
$$\times \psi_{\ell j}(r_1, r_2) \psi_{\ell' j'}(r_1, r_2) Y_{L0}(\theta_{12}), \tag{7}$$

where $\hat{\ell} = \sqrt{2\ell + 1}$ and $\psi_{\ell j}(r_1, r_2)$ is the radial part of the two-particle wave function which is determined from Eq. (3) by making use of Eqs. (5) and (6) of [13].

Figure 4 shows the two-particle density plotted as a function of the radius $r_1 = r_2 = r$ and their opening angle $\theta_{12}$, with a weight factor of $4\pi r^2 \cdot 2\pi r^2 \sin\theta_{12}$ for both Sets A (upper panel) and B (lower panel). The distribution at smaller and larger $\theta_{12}$ are referred to as "di-neutron" and "cigar-like" configurations, respectively. One can see in Fig. 4 that the two-particle density is well concentrated around $\theta_{12} \leq 90°$ for both Sets A (upper panel) and B (lower panel), which is the clear indication of the di-neutron correlation. The di-neutron component has a relatively higher density in comparison to the small cigar-like component for both sets in the ground state of $^{11}$Li. The two peak structure in the two-particle density is attributed to the mixing of the $s$- and $p$-wave components ($\ell \leq 1$) in the ground state of $^{11}$Li.

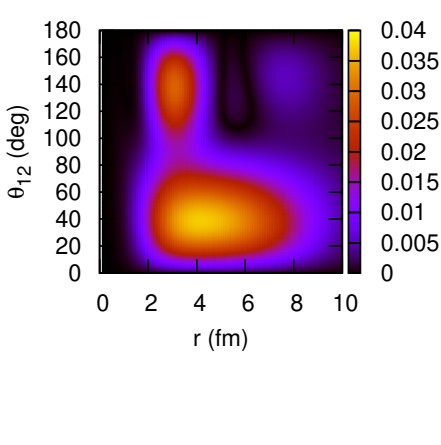

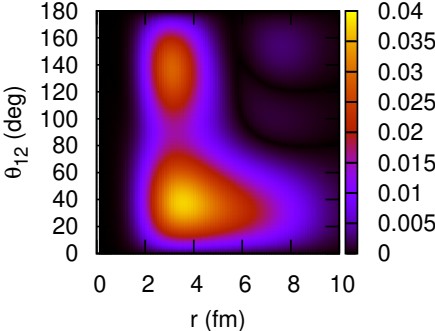

Figure 4: Two-particle density for the ground state of $^{11}$Li for Set A (upper-panel) and Set B (lower-panel) as a function $r_1 = r_2 = r$ and the opening angle between the valence neutrons $\theta_{12}$ for settings mentioned in caption of Table 3 .

## 5   Summary

In the present study we report the emergence of bound $2n$-halo ground state of $^{11}$Li from the coupling of four unbound $spd$-waves in the continuum of $^{10}$Li due to the presence of pairing interaction. The configuration mixing in the ground state of $^{11}$Li has been reported for the two particular choices of core+$n$ potential, fixed in the view of the available recent experimental data. Also, the $2n$-neutron correlation for this system showing prominence of the di-neutron component is discussed. However our results shows different configuration mixing for two different choices of core+$n$ potential. In order to conclude which configuration is likely in the ground state of $^{11}$Li, we need further investigations of the reaction observables that are sensitive to partial wave content of the ground state. Investigations with different choices of pairing interactions and inclusion of spin of core ($^9$Li) are in progress and will be reported elsewhere.

## Acknowledgements

J. Singh gratefully acknowledge the encouragement by Prof. K. Ogata and financial support from the research budget of RCNP theory group to attend the EFB24-2019. Fruitful discussions with Prof. A. Vitturi, Prof. L. Fortunato, Dr. J. Casal and Dr. Y. Kucuk are gratefully acknowledged.

**Funding information** This work was in part supported by JSPS KAKENHI Grants Nos. 18K03635, 18H04569, and 19H05140. W. H. acknowledges the collaborative research program 2019, information initiative center, Hokkaido University.

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
