# Peer review of "Impact of uncertainties of unbound 10Li on the ground state of two-neutron halo 11Li"

_SciPost Physics Proceedings, doi:SciPost Phys. Proc. 3, 007 (2020)_

## Round 1 · Referee Report · Eduardo Garrido (Referee 1) · 2019-12-2

Report
In this paper the authors investigate the properties of $^{11}$Li for two different choices of the $^{10}$Li-n potential.
The main remark I have refers actually to the choice made for the core-neutron potential. The two sets, A and B, use the same $p$-wave and $d$-wave potentials, which have the characteristic of using for both, $p$-waves and $d$-waves, a positive spin-orbit strength. This is reasonable for $p$-waves, since in this way the $p_{3/2}$ states are pushed up in energy, as it should be done to take care in some way of the Pauli principle (the $p_{3/2}$ shell is occupied by the neutrons in the core). However, if the same positive spin-orbit strength is used for the $d$-waves, the consequence is that the $d_{5/2}$ states will be higher than the $d_{3/2}$, which is very likely not right. In the text it is said that the $d_{5/2}$ resonance lies around 2.98 MeV. Since the spin-orbit strength is positive I guess there is a $d_{3/2}$ state below this energy. Is this correct? In any case some comments about this I believe are necessary.
Also, the two potential sets give rise to a quite different partial wave content in $^{11}$Li. Is there any conclusion by the authors concerning which of these two $^{11}$Li structures is more likely? Some comment about this, at least in the conclusions, would be welcome.
The main remark I have refers actually to the choice made for the core-neutron potential. The two sets, A and B, use the same $p$-wave and $d$-wave potentials, which have the characteristic of using for both, $p$-waves and $d$-waves, a positive spin-orbit strength. This is reasonable for $p$-waves, since in this way the $p_{3/2}$ states are pushed up in energy, as it should be done to take care in some way of the Pauli principle (the $p_{3/2}$ shell is occupied by the neutrons in the core). However, if the same positive spin-orbit strength is used for the $d$-waves, the consequence is that the $d_{5/2}$ states will be higher than the $d_{3/2}$, which is very likely not right. In the text it is said that the $d_{5/2}$ resonance lies around 2.98 MeV. Since the spin-orbit strength is positive I guess there is a $d_{3/2}$ state below this energy. Is this correct? In any case some comments about this I believe are necessary.
Also, the two potential sets give rise to a quite different partial wave content in $^{11}$Li. Is there any conclusion by the authors concerning which of these two $^{11}$Li structures is more likely? Some comment about this, at least in the conclusions, would be welcome.

Author: Jagjit Singh on 2019-12-11 [id 672]
(in reply to Report 1 by Eduardo Garrido on 2019-12-02)Dear Referee, Thanks for taking time to read our manuscript. Please find below our reply on your remarks. 1. No, we use the standard spin-orbit interaction, that ensures the normal ordering of the d5/2 and d3/2 states. We will explicitly add "standard choice of spin orbit interaction" to the manuscript in section 3. 2. However our results show different configuration mixing for two different choices of core+n potential. In order to conclude which configuration is likely in the ground state of 11Li, we need further investigations of the reaction observables that are sensitive to partial wave content of the ground state. We will add the same in summary of manuscript.
We will submit revised manuscript soon. Best Regards Jagjit
Anonymous on 2019-12-12 [id 678]
(in reply to Jagjit Singh on 2019-12-11 [id 672])Dear E. Garrido,
Thanks for the question.
In our approach, all continuum single-particle states are orthogonal
to the redundant bound states. Our basis functions are free
from the Pauli forbidden states.
Best Regards
Jagjit
Eduardo Garrido on 2019-12-11 [id 673]
(in reply to Jagjit Singh on 2019-12-11 [id 672])Thank you for the reply. I can now see that a positive $V_{\ell s}$ is the standard spin-orbit interaction due to the additional minus sign that the $d/dr$ is actually producing. But then my question is about the $p$-states, since in this case there should be a $p_{3/2}$ state below the $p_{1/2}$, may be bound, that is Pauli forbidden. How is this taken into account?

---

## Round 3 · Author Response

Following Referee remarks the suggested minor corrections has been
implemented in this version of manuscript.
Best Regards,
Jagjit

---

## Round 3 · List of Changes

Considering the Referee remarks following changes has been made:
1. In sec. 3 we have added "with standard choice of spin-orbit interaction" just before equation 5.
2. In summary we have added
"However our results show different configuration mixing for two different choices of core+n potential. In order to conclude which configuration is likely in the ground state of 11Li, we need further investigations of the reaction observables that are sensitive to partial wave content of the ground state."
3. Minor typo in abstract and third paragraph of intro
d(9Li,p)11Li --> d(9Li,p)10Li

---

## Editorial Decision

published